# Effect of Oral Zinc Supplementation on Phase Angle and Bioelectrical Impedance Vector Analysis in Duchenne Muscular Dystrophy: A Non-Randomized Clinical Trial

**DOI:** 10.3390/nu16193299

**Published:** 2024-09-29

**Authors:** Karina Marques Vermeulen-Serpa, Márcia Marilia Gomes Dantas Lopes, Camila Xavier Alves, Evellyn Camara Grilo, Thais Alves Cunha, Carolinne Thaisa de Oliveira Fernandes Miranda, Breno Gustavo Porfirio Bezerra, Lucia Leite-Lais, José Brandão-Neto, Sancha Helena de Lima Vale

**Affiliations:** 1Postgraduate Program in Health Sciences, Federal University of Rio Grande do Norte, Natal 59012-300, RN, Brazil; karinavermeulen@hotmail.com (K.M.V.-S.); evellyn-cg@hotmail.com (E.C.G.); thaisalvesc1@gmail.com (T.A.C.); brandao-neto@live.com (J.B.-N.); 2Department of Nutrition, Federal University of Rio Grande do Norte, Natal 59078-970, RN, Brazil; marilia.lopes@ufrn.br (M.M.G.D.L.); lucia.leite@ufrn.br (L.L.-L.); 3Nutrition Division, Liga Norte Riograndense Contra o Câncer, Natal 59040-000, RN, Brazil; camila_xavieralves@yahoo.com.br; 4Postgraduate Program in Medicine (Hematology/Oncology), Federal University of São Paulo, São Paulo 04021-001, SP, Brazil; carolt_@hotmail.com; 5Center for Primary Processing and Reuse of Produced Water and Waste, Federal University of Rio Grande do Norte, Natal 59078-970, RN, Brazil; brenogpb@gmail.com; 6Postgraduate Program in Nutrition, Federal University of Rio Grande do Norte, Natal 59078-970, RN, Brazil

**Keywords:** rare diseases, electrical impedance, body composition, micronutrients, pediatrics

## Abstract

Zinc plays a crucial role in cell structure and functionality. Neurodegenerative Duchenne muscular dystrophy (DMD) alters muscle membrane structure, leading to a loss of muscle mass and strength. The objective of this study was to evaluate the changes in phase angle (PA) and bioelectrical impedance vector analysis (BIVA) results in patients with DMD after oral zinc supplementation. This clinical trial included 33 boys aged 5.6 to 24.5 years diagnosed with DMD. They were divided into three groups according to age (G1, G2, and G3) and supplemented with oral zinc. The mean serum zinc concentration was 74 μg/dL, and 29% of patients had concentrations below the reference value. The baseline values (mean (standard deviation)) of the bioelectrical impedance parameters PA, resistance (R), and reactance (Xc) were 2.59° (0.84°), 924.36 (212.31) Ω, and 39.64 (8.41) Ω, respectively. An increase in R and a decrease in PA and lean mass proportional to age were observed, along with a negative correlation (r = −0.614; *p* < 0.001) between age and PA. The average cell mass in G1 was greater than that in G3 (*p* = 0.012). There were no significant differences in serum zinc levels or bioelectrical impedance parameters before and after zinc supplementation. We conclude that this population is at risk of zinc deficiency and the proposed dosage of zinc supplementation was not sufficient to alter serum zinc levels, PA and BIVA results.

## 1. Introduction

Zinc is one of the most abundant micronutrients in the human body, known for its regulatory, structural, and catalytic role in at least 3000 proteins, including enzymes and transcription factors [1]. This micronutrient plays an important role in the integrity and stabilization of structural membranes and cellular functionality [2].

The participation of zinc in membrane stability is described in the literature through three mechanisms: (a) zinc promotes the association between membrane proteins and cytoskeletal proteins; (b) zinc stabilizes the reduced form of sulfhydryl groups, contributing to antioxidant protection against the effects of membrane disruption caused by lipid and protein oxidation; (c) zinc preserves the integrity of ion channels, thus acting as an antagonist to the adverse effect of free Ca^2+^ [3,4].

Efficient homeostatic systems regulate intracellular Zn concentrations. However, failure in these systems has been implicated in the pathophysiology of several human diseases, from cancer to neurological and neuropsychiatric disorders, diabetes and infections [1]. Additionally, its deficiency increases the fragility of the erythrocyte membrane, compromising platelet aggregation, osmotic function, and several other processes [5,6].

The global prevalence of zinc deficiency is estimated to range between 17% and 20%, with the vast majority occurring in developing countries such as Africa and Asia [3].

Zinc supplementation in healthy children promotes a positive effect on cellular integrity and functionality, as demonstrated by an increase in the phase angle (PA) value and an upward shift in the bioelectrical impedance vector [7]. However, to date, there is a lack of studies involving zinc supplementation and its effect on PA and bioelectrical impedance vector analysis (BIVA) in neurogenerative diseases.

PA is related to cellular integrity and functionality as well as lean mass. Therefore, it has been used for nutritional evaluation and as a prognostic parameter in several clinical situations [8]. Diseases, inflammation, malnutrition, and prolonged physical inactivity negatively affect the electrical properties of tissues, resulting in PA values lower than those of healthy individuals [9]. BIVA is based exclusively on the electrical properties of tissues and is determined by resistance (R) and reactance (Xc) values. Consequently, BIVA can be used to monitor changes in tissue hydration (represented by R) and structure (represented by Xc). This method has been demonstrated to be effective in detecting changes in hydration or body composition for clinical conditions where traditional assessment methods are unreliable [10].

Duchenne muscular dystrophy (DMD) is a severe degenerative neuromuscular disease that affects 1 in 5000 live births [11]. This disease is caused by mutations in the DMD gene (Xp21.2), which encodes the dystrophin protein. Reduced levels or absence of this protein cause changes in the structure of the muscle plasma membrane, resulting in progressive loss of lean mass and muscle strength. This condition leads to muscle degeneration and necrosis [12,13], ultimately resulting in severe disability and death during late adolescence [14].

Disease progression and side effects of glucocorticoid therapy strongly impair the nutritional status of DMD patients. As children are affected by the disease progression, it becomes crucial to address nutritional aspects, such as the transition from lean mass to fat mass, the harmful impact of malnutrition on glucose metabolism, as well as respiratory and cardiac functions [15,16,17]. Additionally, anthropometric measurement techniques and reference values used in the general population of the same age group are not applicable to DMD patients due to inherent changes in growth and body composition caused by the disease. This hinders nutritional diagnosis and clinical-nutritional follow-up in these patients [15].

Considering the role of zinc in the body and the existing knowledge of the pathophysiology of DMD, along with the lack of studies on this topic, this study aimed to evaluate the changes in PA and BIVA results in DMD patients after oral zinc supplementation.

## 2. Materials and Methods

### 2.1. Study Population

This non-randomized clinical trial was conducted after receiving approval from the Ethics and Research Committee in Humans (1.754.017) and was registered in the Brazilian Registry of Clinical Trials (RBR-7cfdxm). All participants and/or their legal guardians signed consent forms authorizing their participation in the study.

Forty-five patients with a clinical history of DMD were recruited between February 2018 and March 2020. The patients were treated at the Onofre Lopes University Hospital (HUOL) in Natal, Brazil. Participants aged 5 years and older with a confirmed genetic diagnosis of DMD were included.

Participants were allocated into three groups (Figure 1) following the age groups established by the Institute of Medicine [18] for defining zinc daily requirements: group 1 (G1; age, 5–8 years) with an estimated average requirement (EAR) of 4.0 mg; group 2 (G2; age, 9–13 years) with an EAR of 7.0 mg; and group 3 (G3; age ≥ 14 years) with an EAR of 8.5 mg. Data were collected in two phases (T1 and T2), with an interval of four months between them. In both phases, patients underwent a medical history review, physical examination, serum zinc analysis, anthropometric assessment, and bioelectrical impedance analysis (BIA). Participants who could not undergo BIA (*n* = 9) and those who did not complete the zinc supplementation regimen (*n* = 2) were excluded from the analysis.

### 2.2. Intervention: Oral Zinc Supplementation

Participants in all groups were prescribed an oral supplement containing Taste-FreeTM Zinc (zinc bisglycinate chelate, unflavored; Albion Laboratories, Layton, UT, USA) to be taken daily for four months. Zinc supplementation dose was based on previous data regarding zinc consumption by healthy children aged 6 to 9 years living in Brazil [19]. The dosage was established to ensure that the total amount consumed and supplemented did not exceed the upper intake levels established by the Institute of Medicine [18].

The established doses for G1, G2, and G3 were 5, 10, and 15 mg/day, respectively. A total of 120 zinc doses were distributed to each participant. The doses were delivered at three different times over four months, packaged in sets of 40 doses each, and administered in the home environment. Researchers conducted weekly phone calls with the participants’ legal guardians to monitor supplementation.

### 2.3. Serum Zinc Assessment

From each participant, 4 mL of blood was collected for serum zinc analysis. Blood was collected in vacuum tubes containing a separator gel and clot activator. After separation, the serum was stored at −80 °C until analysis. All collection and storage procedures were performed as described by the International Zinc Nutrition Consultative Group (IZiNCG) (Oakland, CA, USA) [20].

Among the 33 serum samples, 28 were analyzed, and 5 were discarded due to hemolysis. Zinc concentrations were analyzed using an inductively coupled plasma atomic emission spectrometer (ICP-OES) model iCAP 6300 Duo with a simultaneous Charge Injection Detector (CID) (Thermo Fisher Scientific, Bremen, Germany).

The reference values for serum zinc used were 65 μg/dL and 70 μg/dL for participants aged <10 years and >10 years, respectively [20]. To assess the risk of zinc deficiency in this population, the IZiNCG recommends considering the risk of zinc deficiency in the population (or subgroup) where more than 20% of individuals (or a population subgroup) have serum zinc concentrations below the cutoff point [20].

### 2.4. Anthropometric Assessment

A platform scale (KN Waagen, São Paulo, Brazil) was used to obtain weight measurements (kg), providing a ramp for patients in wheelchairs. Height (m) was measured using a stadiometer (Sanny, São Paulo, Brazil) or estimated using segmented recumbent height measured with an inextensible millimeter tape.

Body mass index (BMI) was calculated as the ratio of body weight to height squared (kg/m^2^). Weight-for-age (WAZ), height-for-age (HAZ), and BMI-for-age (BAZ) Z-score values were determined using the software AnthroPlus v1.0.4 (available at www.who.int/growthref/en/ (accessed on 10 September 2023). These parameters were classified according to the World Health Organization growth curves for healthy individuals aged between 2 and 19 years [21]. Nutritionists who had undergone training and calibration performed the anthropometric assessments.

### 2.5. Bioelectrical Impedance Parameters

Bioelectrical impedance parameters, R (Ω) and Xc (Ω), were obtained using a Quantum II^®^ tetrapolar body composition analyzer (RJL Systems, Clinton Township, MI, USA) with a single, safe, and painless electrical frequency (50 kHz). Two self-adhesive electrodes were placed on the dorsal surface of the right hand and two on the dorsal surface of the right foot of the participant in the supine position, as described by Lukaski et al. [22]. The measured R and Xc parameters were used to determine the percentage of fat-free mass (FFM), PA, and for BIVA.

FFM (%) was calculated using the equation proposed by Houtkooper et al. [23]. PA was calculated using the formula PA = arc tang (Xc/R) [24]. The PA reference value of ≥5.79° was adopted based on literature for Brazilian children [25].

BIVA results were based on normalized R and Xc values for height measurements (R/H and Xc/H in Ω/m). In the RXc graph, the standardized impedance measurements of the children were represented as bivariate vectors with confidence and tolerance intervals, depicted as ellipses on the RXc plane. Confidence ellipses were determined using BIVA 2002 software developed by Piccoli et al. [26]

To investigate the differences between groups, we plotted the 95% confidence intervals for the mean impedance difference of the bivariate vectors measured before and after supplementation for each group. The position and length of the vector provide information regarding hydration status, cell mass, and cell integrity. An upward displacement of the ellipse from the main axis is associated with a larger cell mass, while a downward displacement is associated with a smaller cell mass. The significant value for the T2 statistic indicates that the mean vectors of each group analyzed are different [26].

### 2.6. Statistical Analysis

Statistical analysis involved assessing data distribution using the Shapiro–Wilk test. Quantitative variables with a normal distribution were presented as mean and standard deviation (SD) values, while variables deviating from the Gaussian model were presented as median values with lower and upper limits. Student’s *t*-test and Wilcoxon test were used to compare variables (anthropometric and bioelectrical impedance variables and serum zinc level) between T1 and T2, depending on the normality of the data distribution. Pearson’s correlation test was used to analyze the relationship between age and PA. Hoteling T2 test was performed using the BIVA software (2002). Statistical results were considered significant at 5% (*p* ≤ 0.05).

## 3. Results

This study included 33 male patients with DMD (age range, 5.6–24.5 years; median age, 12 years). DMD genetic tests revealed exonic deletions in 64% of patients and base duplications, insertions, or substitutions in the remaining patients. Regarding clinical status and nutritional support, none of the participants had acute illnesses, 88% were prescribed oral glucocorticoids, and 85% were receiving oral calcium and vitamin D supplementation as part of their outpatient treatment.

The initial mean serum zinc concentration of the assessed population was 74.0 (12.0) μg/dL. The study population was found to be at risk of zinc deficiency since 29% of patients had serum zinc levels below the reference value. The percentage of zinc deficiency was consistent across all groups (G1, 29%; G2, 27%; G3, 30%).

Zinc supplementation numerically increased the mean serum zinc in all groups, but this result was not statistically significant. Regarding the eight boys who had serum zinc levels below the reference value, three of them had their values restored to normal after supplementation.

Overall, the mean (SD) values of bioelectrical impedance parameters PA, R, and Xc were 2.59° (0.84°), 924.36 (212.31) Ω, and 39.64 (8.41) Ω, respectively. As shown in Table 1, an increase in R values and a decrease in PA and FFM values were observed proportionally with age. In addition, the mean BAZ decreased. There were no significant differences in bioelectrical impedance parameters before and after zinc supplementation.

Furthermore, a negative correlation (r = −0.614; *p* < 0.001) was observed between age and PA in the overall population. Another correlation between PA and bioelectrical impedance parameters was tested, but no significant relationship was found. Regarding BIVA, when comparing the groups at T1 (Figure 2), G1 had a significantly larger mean cell mass compared with G3 (*p* = 0.012). There were no significant differences in cell mass between the time points (Figure 3).

## 4. Discussion

This original study evaluated the serum zinc levels, anthropometric, and bioelectrical impedance parameters in 33 male patients with DMD who received oral zinc supplementation for four months. Changes in PA and BIVA were observed even before detectable changes in anthropometric measurements. Serum zinc levels did not change with oral zinc supplementation.

The oral zinc supplement was offered in the form of zinc bisglycinate chelate, which has high absorption and bioavailability. Most patients received vitamin D supplementation as part of their outpatient treatment. It is known that zinc is an essential cofactor for the functions of vitamin D, in the same way that vitamin D can also positively influence the absorption and homeostasis of zinc by regulating its transporters [27].

Studies assessing zinc deficiency in DMD patients are scarce, and these studies have limited sample sizes [28,29]. At the beginning of the present study, 29% of the patients had serum zinc levels lower than the reference value. After oral zinc supplementation, we found a decrease (to 20%) in the percentage of deficiency in this population, but we did not find a significant increase in the serum zinc value, indicating the population remains at risk of zinc deficiency.

Two factors may explain this result. First, it might be due to low consumption of food sources of zinc, as reported by Bernabe-García et al. [30], who analyzed the food consumption of 101 boys with DMD and found high energy intake but insufficient zinc intake. Additionally, other studies have shown that dietary patterns among children with DMD are inadequate, with nutritional deficiencies coexisting alongside overnutrition [31].

Secondly, it is well-documented in the literature that serum zinc levels can be influenced by several factors such as lean body mass, body fat mass, oxidative stress, and infections [32]. Thus, in DMD patients, serum zinc levels may be low due to increased inflammation and oxidative stress, which are characteristics of the disease, as well as a decrease in lean mass, as observed in the present study.

In addition to the decrease in FFM (%), PA decreases with advancing age, showing values much lower than the reference values [25]. Low PA values are associated with a smaller body cell mass and cellular water imbalance. During inflammation, Xc decreases in response to the lower capacitance of damaged cell membranes. This occurs concurrently with a reduction in R due to decreased intracellular activity and an increased volume of extracellular water volume. Since PA represents the ratio between Xc and R, a greater decrease in Xc compared to R during inflammation results in a decrease in PA. Therefore, a low PA value is directly related to muscle mass loss, disease progression, increased inflammation, poor quality of life, malnutrition, and increased mortality [33,34,35].

In a recent study by Tsuji et al. [36], the correlation between sarcopenia and PA was investigated, suggesting that PA might be a valid discriminator of sarcopenia in patients with chronic musculoskeletal pain. Furthermore, performing localized bioimpedance analysis in patients with neuromuscular diseases, Rutkove et al. [37] reported that reductions and normalization of PA were correlated to disease progression and remission, respectively.

In this study, BIVA results reaffirmed that the older the age group, the smaller the cell mass, as indicated by the upward displacement of the ellipse in G1 compared to G3. BIVA reflects variations in bioelectrical patterns and enables monitoring of changes in nutritional status and body composition evaluation. Thus, BIVA is recommended, especially when calculating body composition is not possible, such as in patients with DMD [38].

A previous study [7] involving 71 healthy eutrophic children reported that oral zinc supplementation at 10 mg/day promoted changes in serum zinc, Xc and PA, and vector displacement in BIVA. These changes were associated with improvements in cell membrane integrity and increased cell mass. Zinc plays a crucial role in the integrity, stabilization, protection, and functionality of structural membranes [2,39]. However, in the present study, we were unable to confirm this as zinc supplementation also had no effect on the PA and BIVA parameters.

DMD is a progressive neurodegenerative disease that induces permanent changes in muscle cell membranes [11]. Therefore, zinc supplementation may not be enough to alter or improve the measured parameters.

Another reflection from this study is that the values recommended by the Institute of Medicine [18] for healthy children may not be adequate for patients with DMD. Given inflammation and lean mass loss in DMD, doses higher than those recommended for zinc are likely necessary to achieve an effect on the measured parameters.

Our study’s strengths include conducting an innovative clinical trial with zinc supplementation using non-standardized tools for nutritional assessment of DMD patients. Our sampling procedure was limited because it was based on a non-probability method. However, we included all patients treated in our clinic during the described period, dividing them into age-based groups. Moreover, due to the experimental nature of this study, we cannot ensure that all patients adhered to the correct daily intake of the supplements. To address potential issues, we implemented packaging changes and conducted weekly communication sessions with participants’ legal guardians to monitor supplement administration.

The generalizability of our findings should be evaluated in future studies with more representative samples and conducted over a longer period with higher levels of oral zinc supplementation.

## 5. Conclusions

We found that boys with DMD were at risk for zinc deficiency. Nevertheless, oral zinc supplementation in the amount offered did not significantly change serum zinc, PA, and BIVA parameters.

Considering the functions of zinc and the pathophysiology of DMD, it is essential to implement strategies to prevent or treat zinc deficiency in this population. Further studies investigating larger and safer doses of oral zinc supplementation are warranted.

Furthermore, our findings suggest that the assessment of PA and BIVA is useful for clinical-nutritional evaluation and monitoring of DMD patients.

## Figures and Tables

**Figure 1 nutrients-16-03299-f001:**
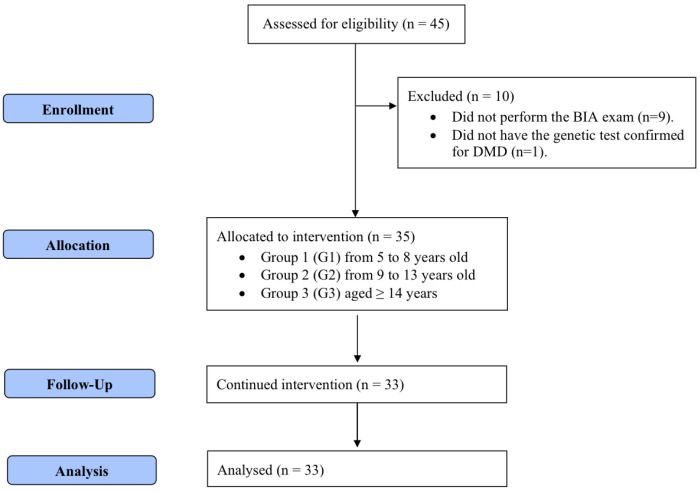
CONSORT flowchart for recruitment, selection, and analysis of participants.

**Figure 2 nutrients-16-03299-f002:**
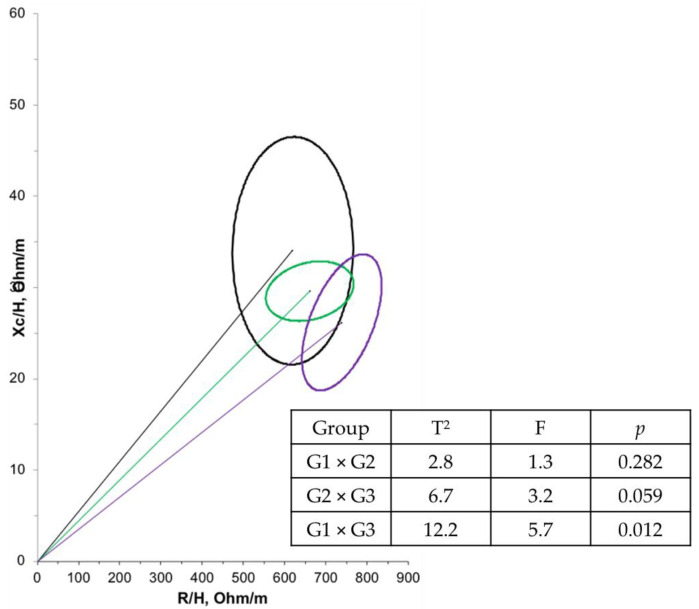
Confidence ellipses of 95% of impedance vectors measured before (T1) intervention with different groups (G1: black ellipse; G2: green ellipse; G3: purple ellipse).

**Figure 3 nutrients-16-03299-f003:**
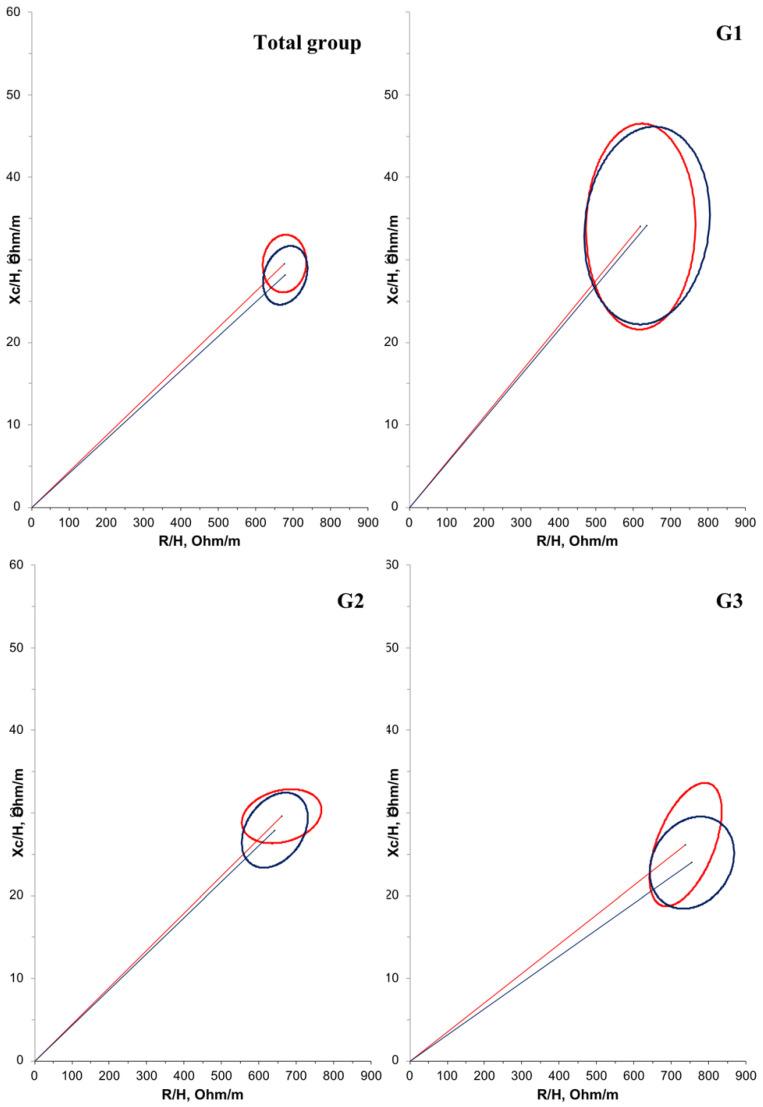
Confidence ellipses of 95% of impedance vectors measured before (T1: red ellipse) and after (T2: blue ellipse) oral zinc supplementation in DMD patients. The upward or downward displacement of the main axis is associated with larger or smaller cell mass, respectively.

**Table 1 nutrients-16-03299-t001:** Serum zinc and bioelectric parameters before (T1) and after (T2) oral zinc supplementation in DMD patients by age group.

Groups	Parameters	T1	T2	*p*
G1 (*n* = 8)	Serum zinc (μg/dL) *	68 (58; 86)	75 (62; 95)	0.599
	WAZ *	−0.64 (−1.32; 0.51)	−0.88 (−1.48; 0.62)	–
	HAZ *	−1.66 (−2.48; −0.57)	−1.80 (−2.73; −0.49)	–
	BAZ *	0.46 (−0.22; 1.88)	0.42 (−0.46; 2.09)	–
	FFM (%) *	80.8 (68.8; 87.8)	81.2 (66.6; 89.3)	–
	R (Ω) *	645.5 (595.5; 810.7)	650.5 (607.7; 856.3)	0.093
	Xc (Ω) *	35.5 (30.93; 45.6)	36.0 (31.7; 46.3)	0.156
	PA (°) *	3.22 (2.36; 4.11)	3.05 (2.28; 4.12)	0.262
G2 (*n* = 14)	Serum zinc (μg/dL)	75 (11.0)	80 (10.0)	0.466
	HAZ	−0.81 (1.44)	−0.92 (1.43)	–
	BAZ	0.25 (2.86)	0.29 (2.79)	–
	FFM (%)	57.4 (13.5)	57.5 (14.6)	–
	R (Ω)	919.5 (174.2)	904.3 (139.6)	0.508
	Xc (Ω)	41.3 (5.70)	39.4 (7.70)	0.234
	PA (°)	2.67 (0.64)	2.58 (0.71)	0.181
G3 (*n* = 11)	Serum zinc (μg/dL)	74 (13.0)	78 (22.0)	0.459
	HAZ	−2.60 (0.60)	−2.81 (0.71)	–
	BAZ	−1.40 (3.40)	−1.35 (3.29)	–
	FFM (%)	54.4 (17.36)	54.0 (16.60)	–
	R (Ω)	1091 (152)	1119 (157)	0.401
	Xc (Ω)	38.5 (11.1)	35.6 (8.72)	0.553
	PA (°)	2.02 (0.51)	1.85 (0.52)	0.490

WAZ, weight-for-age Z-score; HAZ, height-for-age Z-score; BAZ, BMI-for-age Z-score; FFM, fat-free mass; R, resistance; Xc, reactance; PA, phase angle. * Data with non-parametric distribution; Data are presented as mean (standard deviation) or median (95% CI, lower; upper values).

## Data Availability

The data presented in this study are available on request from the corresponding author due to ethical reasons and participant privacy.

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
