# Peer review of "Effect of Oral Zinc Supplementation on Phase Angle and Bioelectrical Impedance Vector Analysis in Duchenne Muscular Dystrophy: A Non-Randomized Clinical Trial"

_nutrients, 2024, doi:10.3390/nu16193299_

Round 1

Reviewer 1 Report

Comments and Suggestions for Authors

This is a quite interesting study concerning effect of zinc supplementation on some parameters in patients with DMD) measured by bioelectrical impedance. Introduction justifies the study; however, authors could give additional information about phase angle, what's this and why it is important to be analyzed. Methods which were applied are appropriate. Results are almost clearly presented. Something's wrong in case of serum zinc data - minimum and maximum values for G1 group don't match to the median value. As for G2 and G3 groups - are you sure about so small standard deviation? Please check your data again.

You should give more information about these 29% of patients who had serum zinc levels lower than the reference value. How about their levels after supplementation? For the clarity of your paper, it would be nice if you presented this reference values.

Discussion and conclusions are fair enough, however, it would be fair to write about weakness of your study, for example are you sure that your patients really took the recommended  supplements?

Besides:

Line 46 - What’s AF? You need to explain all abbreviations.

Author Response

Comments 1: This is a quite interesting study concerning effect of zinc supplementation on some parameters in patients with DMD) measured by bioelectrical impedance. Introduction justifies the study; however, authors could give additional information about phase angle, what's this and why it is important to be analyzed. Methods which were applied are appropriate.

Response 1: We are grateful for your opinion, and we agree that the Introduction Section could be more informative about phase angle and BIVA. We added the following information (lines 60-69):

PA is related to cellular integrity and functionality as well as lean mass. Therefore, it has been used for nutritional evaluation and as a prognostic parameter in several clinical situations. Diseases, inflammation, malnutrition, and prolonged physical inactivity negatively affect the electrical properties of tissues, resulting in PA values lower than those of healthy individuals. BIVA is based exclusively on the electrical properties of tissues and is determined by resistance (R) and reactance (Xc) values. Consequently, BIVA can be used to monitor changes in tissue hydration (represented by R) and structure (represented by Xc). This method has been demonstrated to be effective in detecting changes in hydration or body composition for clinical conditions where traditional assessment methods are unreliable.

Comments 2: Results are almost clearly presented. Something's wrong in case of serum zinc data - minimum and maximum values for G1 group don't match to the median value. As for G2 and G3 groups - are you sure about so small standard deviation? Please check your data again.

Response 2: Thank you for your attention at this point. We revised and noticed there was a mistake, and the values of standard deviation and minimum and maximum presented were in mg/L, so we changed for μg/dL as the other values and the literature references.  

Comments 3: You should give more information about these 29% of patients who had serum zinc levels lower than the reference value. How about their levels after supplementation? For the clarity of your paper, it would be nice if you presented this reference values.

Response 3: Thank you for your considerations. We rewrote this part of the Results (lines 196 - 199) and Discussion (lines 229 - 232) sections to make this information clearer and added more information about these groups. To assess the risk of zinc deficiency in this population, the IZiNCG recommends considering the risk of zinc deficiency in the population (or subgroup) where more than 20% of individuals (or a population subgroup) have serum zinc concentrations below the cutoff point. This statement was previously described in lines 135-138. 

Comments 4: Discussion and conclusions are fair enough, however, it would be fair to write about weakness of your study, for example are you sure that your patients really took the recommended  supplements?

Response 4: Thank you for your thoughtfulness. We rewrote the weaknesses of the study to make it clearer and added this information (lines 294–298).

Comments 5: Line 46 - What’s AF? You need to explain all abbreviations.

Response 5: Thank you for your attention at this point. This was a mistake; the correct form in line 58 is PA (phase angle). 

Reviewer 2 Report

Comments and Suggestions for Authors

Manuscript written by Karina Vermeulen-Serpa et al. presents the results of an interesting study, the aim of which was evaluation of the changes in phase angle and bioelectricasl impedence vector analysis results in patients with Dunchenne muscular dytrophy after zinc supplementation.

While reading and analyzing the manuscript, I decided to propose the following suggestions and ask quentions to the authors:

1.       Lines: 36-37: Please provide the molecular mechanism that indicates that zinc plays an important role in the integrity and stabilization of structural membranes and cellular functinality.

2.       Lines: 40-42: Please expand on what zinc deficiency can lead to. Can it contribute to the development of diseases?

3.       What do you think about conducting an additional correlation between zinc levels and R, Xc, PA separately before (T1) and after (T2) zinc supplementation?

4.       At the end of the Discussion section, please write about further development prospects for this study.

5.       Have you wondered what can increase zinc absorption? Have you considered supplementing zinc with another component to increase the effects of zinc?

6.       What do you think about zinc supplementation in combination with vitamin D?

Author Response

Comments 1: Lines: 36-37: Please provide the molecular mechanism that indicates that zinc plays an important role in the integrity and stabilization of structural membranes and cellular functinality.

Lines: 40-42: Please expand on what zinc deficiency can lead to. Can it contribute to the development of diseases?

Response 1: We appreciate your time and consideration. These information were inserted in lines 40 - 49.

Comments 2: What do you think about conducting an additional correlation between zinc levels and R, Xc, PA separately before (T1) and after (T2) zinc supplementation?

Response 2: We examined these data but did not find statistical results. We added this information in lines 206 - 207.

Comments 3: At the end of the Discussion section, please write about further development prospects for this study.

Response 3: We are grateful for this insight. We added this statement in lines 301 - 303: 

The generalizability of our findings should be evaluated in future studies with more representative samples and conducted over a longer period with higher levels of oral zinc supplementation. 

Comments 4: Have you wondered what can increase zinc absorption? Have you considered supplementing zinc with another component to increase the effects of zinc?

What do you think about zinc supplementation in combination with vitamin D?

Response 4: The oral zinc supplement was offered in the form of zinc bisglycinate chelate, which has high absorption and bioavailability. Most patients received vitamin D supplementation as part of their outpatient treatment. It is known that zinc is an essential cofactor for the functions of vitamin D, in the same way that vitamin D can also positively influence the absorption and homeostasis of zinc by regulating its transporters. We added this information in lines 230 - 235.

Reviewer 3 Report

Comments and Suggestions for Authors

The article is thorough and detailed, examining the effects of zinc supplementation in patients with Duchenne muscular dystrophy. The results and conclusions are well-considered, supported by comprehensive statistics, and provide valuable insights into the issue of zinc deficiency related to DMD. It would be worthwhile to investigate whether similar results would be obtained with a larger sample size. Additionally, conducting a longer-term study and exploring how higher doses or other supplementary therapies might affect serum zinc levels and bioelectrical parameters would be beneficial. Further elaboration on how zinc affects muscle cells at the cellular level (a more detailed description of the mechanism of action), especially in the case of DMD, would be valuable. What molecular or cellular mechanisms might be at play? To what extent could dietary modifications have supplemented zinc intake (what specific foods and in what amounts? Are there existing studies on this?). Overall, the article is well-founded and informative, but further research and more detailed analyses are necessary to gain a complete understanding of the role of zinc in the treatment of DMD.

Author Response

Comments 1: The article is thorough and detailed, examining the effects of zinc supplementation in patients with Duchenne muscular dystrophy. The results and conclusions are well-considered, supported by comprehensive statistics, and provide valuable insights into the issue of zinc deficiency related to DMD. It would be worthwhile to investigate whether similar results would be obtained with a larger sample size. Additionally, conducting a longer-term study and exploring how higher doses or other supplementary therapies might affect serum zinc levels and bioelectrical parameters would be beneficial. Further elaboration on how zinc affects muscle cells at the cellular level (a more detailed description of the mechanism of action), especially in the case of DMD, would be valuable. What molecular or cellular mechanisms might be at play? To what extent could dietary modifications have supplemented zinc intake (what specific foods and in what amounts? Are there existing studies on this?). Overall, the article is well-founded and informative, but further research and more detailed analyses are necessary to gain a complete understanding of the role of zinc in the treatment of DMD.

Response 1: I appreciate your words of support and feedback. We acknowledge that conducting research with this particular population is inherently challenging, mainly due to the rarity of the disease and the absence of prior studies examining the effects of nutrient supplementation. This was our first study with lower doses, based on the previous model described in healthy children. In light of these findings, we intend to conduct further investigations involving increased dosages and extended periods of zinc supplementation. This will provide a more comprehensive understanding of the impact of zinc on muscle cells at the cellular level in individuals with DMD.